# Greenspace, Inflammation, Cardiovascular Health, and Cancer: A Review and Conceptual Framework for Greenspace in Cardio-Oncology Research

**DOI:** 10.3390/ijerph19042426

**Published:** 2022-02-19

**Authors:** Jean C. Bikomeye, Andreas M. Beyer, Jamila L. Kwarteng, Kirsten M. M. Beyer

**Affiliations:** 1Institute for Health and Equity, Medical College of Wisconsin, 8701 Watertown Plank Rd., Milwaukee, WI 53226, USA; jbikomeye@mcw.edu (J.C.B.); jkwarteng@mcw.edu (J.L.K.); 2PhD Program in Public and Community Health, Division of Epidemiology & Social Sciences, Institute for Health and Equity, Medical College of Wisconsin, 8701 Watertown Plank Rd., Milwaukee, WI 53226, USA; 3Department of Medicine, Division of Cardiology, Cardiovascular and Cancer Center, Medical College of Wisconsin, 8701 Watertown Plank Rd., Milwaukee, WI 53226, USA; 4MCW Cancer Center, Medical College of Wisconsin, 8701 Watertown Plank Rd., Milwaukee, WI 53226, USA

**Keywords:** greenspace, inflammation, cancer, cardiovascular disease (CVD), cardio-oncology, cancer treatment-related cardiotoxicity, cancer survivors, conceptual framework, socioecological model of health, major adverse cardiovascular events (MACEs)

## Abstract

Cardiovascular disease (CVD) is a leading cause of global morbidity and mortality. Cancer survivors have significantly elevated risk of poor cardiovascular (CV) health outcomes due to close co-morbid linkages and shared risk factors between CVD and cancer, as well as adverse effects of cancer treatment-related cardiotoxicity. CVD and cancer-related outcomes are exacerbated by increased risk of inflammation. Results from different pharmacological interventions aimed at reducing inflammation and risk of major adverse cardiovascular events (MACEs) have been largely mixed to date. Greenspaces have been shown to reduce inflammation and have been associated with CV health benefits, including reduced CVD behavioral risk factors and overall improvement in CV outcomes. Greenspace may, thus, serve to alleviate the CVD burden among cancer survivors. To understand pathways through which greenspace can prevent or reduce adverse CV outcomes among cancer survivors, we review the state of knowledge on associations among inflammation, CVD, cancer, and existing pharmacological interventions. We then discuss greenspace benefits for CV health from ecological to multilevel studies and a few existing experimental studies. Furthermore, we review the relationship between greenspace and inflammation, and we highlight forest bathing in Asian-based studies while presenting existing research gaps in the US literature. Then, we use the socioecological model of health to present an expanded conceptual framework to help fill this US literature gap. Lastly, we present a way forward, including implications for translational science and a brief discussion on necessities for virtual nature and/or exposure to nature images due to the increasing human–nature disconnect; we also offer guidance for greenspace research in cardio-oncology to improve CV health outcomes among cancer survivors.

## 1. Introduction

### 1.1. Cardiovascular Disease Burden

Cardiovascular disease (CVD) is the leading cause of global morbidity and mortality, and it is a major economic burden on healthcare systems [1,2]. In 2016 alone, CVD was the most common underlying cause of global death, accounting for an estimated 17.3 out of 54 million total deaths, or 31.5% of all global deaths [3,4]. Myocardial infarction (MI) and stroke were responsible for 85% of those 31.5% of global deaths [3]. In the 2016 burden of disease study in Europe, CVD was the main leading cause of death, responsible for 45% of all deaths [5]. In the US, CVD remains the number one cause of death [4], followed by cancer [6]. The American Heart Association (AHA) projections indicate that, by 2030, 43.9% of the US adult population will have some form of CVD [4]. CVD is expected to cost the US economy 1.1 trillion USD (about 3400 USD per person in the US) in 2035 [7]. Cancer survivors have significantly elevated CVD risk and are more likely to die from its adverse consequences compared to the general population [8]. In addition, some cancer treatment-related cardiotoxicity amplifies survivors’ risk of major adverse cardiovascular (CV) events (MACEs) [9,10,11], affecting 3.4% to 19.1% of survivors [12]; scholars have also linked inflammation to MACE pathogenesis [13].

Different pharmacological interventions aimed at reducing both inflammation and risk of MACEs have been conducted, but results have been largely mixed to date [14,15,16]. With a growing literature suggesting protective effects of environmental factors such as greenspaces against both CVD and cancer [17], it is necessary to incorporate greenspace-related behavioral interventions in the overall CVD and cancer care process. To help guide that strategic decision-making process, this paper aims to review the state of knowledge on associations among CVD, cancer, inflammation, and previous pharmacological interventions aimed at reducing both inflammation and risk of MACEs, namely, the CV Inflammation Reduction Trial (CIRT), the Colchicine CV Outcomes Trial (COLCOT), and the Canakinumab Anti-Inflammatory Thrombosis Outcomes Study (CANTOS) [14,15,18]. We then discuss the premise of greenspace or nature-based interventions as a potential population-based strategy in the cardio-oncology care continuum that can be added to current clinical or pharmacological interventions in CVD, cancer, and inflammation treatments and management. Lastly, we present an expanded conceptual model to guide greenspace research in cardio-oncology to reduce cancer treatment-induced cardiotoxicity and improve survivorship quality and survival.

### 1.2. Cardiovascular Disease Burden among Cancer Survivors

CVD and cancer are closely linked through common risk factors such as age, tobacco use, poor diet, obesity, psychosocial stress, and sedentary lifestyle [19,20], coexistence of both diseases [9,20,21,22], and increased deleterious effects of cancer treatments on CV systems [20,23,24]. Additionally, recent evidence suggests a bidirectional relationship between cancer and heart failure (HF) [9]. Previous CV events have been shown to promote cancer proliferation [25]. Vice versa, patients with HF have increased risk of incident cancer [26], and the cancer risk is even higher if HF occurs after MI [27]. The close comorbid linkage of CVD and cancer is illustrated by high CVD prevalence among long-term cancer survivors compared to the general population [28]. For example, breast cancer (BC) survivors have increased CVD risk-related deaths [8], particularly if they have pre-existing CVD risk factors [29]. BC survivors are also more likely to die from CVD than cancer recurrence [28]. A population-based case–control study found that the risk of death from CVD among BC survivors was 80% greater compared to age-matched women without BC [8]. Additionally, there is an increased long-term burden of CVD for young adult childhood cancer survivors (CCSs) [30]. CV events increase the risk of death among CCSs and are responsible for a sevenfold higher risk of death in CCSs compared to their age-matched controls [31].

The pathophysiology of cancer treatment-induced cardiotoxicity stems from the design of some chemotherapeutic agents including some antitumor antibiotics such as anthracyclines, which are intended to interfere with cancerous and rapidly dividing cells, the molecular mechanisms of which are reviewed elsewhere [32,33,34]. Unfortunately, while those drugs are effective in cancer treatment, they do not differentiate between cell types and simply kill cells at different specificities, usually more cancer cells than host cells. These drugs have many side-effects, mostly on nondividing cells, including cellular damage [32] and cardiac dysfunction [35]; however, their target mechanism of topoisomerase II does not explain the side-effects as cardiomyocytes hardly divide [36,37]. Cardiomyocytes have limited regenerative capability, which increases their susceptibility to long-term adverse effects from cancer treatment [38]. Anthracycline-related cardiotoxicity, mostly studied as damage to the cardiac muscle itself, manifests as acute HF or subclinical left-ventricular dysfunction which slowly progresses to HF over the course of some years after treatment [39]. 

In addition to anthracyclines, other cancer treatment agents have been associated with cardiomyopathy [9] including alkylating agents that cause endothelial and myocyte damage [40]. Alkylating agent cardiotoxicity is predominantly manifested through pericarditis [9], with high doses leading to myocarditis and HF [41]. Other examples are antimetabolites such as 5-fluorouracil, whose cardiotoxic effects trigger coronary artery vasospasm [42]. Preclinical studies indicate that anthracyclines and 5-fluorouracil trigger reactive oxygen species and induce mitochondrial dysfunction, which make them toxic to surrounding cells, including endothelial cells and cardiac myocytes, leading to progression of arterial stiffness, fibrosis, and other complications [43,44,45].

Additionally, most recent preclinical studies showed cardiotoxic effects of novel targeted cancer therapies [46]. For example, therapies targeting the vascular endothelial growth factor, an angiogenic factor essential in tumor angiogenesis [47], have been linked to many CV side-effects, including hypertension (HTN), thromboembolism, and cardiomyopathy [48,49]. Similarly, during the first pivotal study of trastuzumab, a monoclonal antibody targeting human epidermal growth factor receptor 2 (HER2), symptomatic HF or asymptomatic cardiac dysfunction developed in 27% of patients who received the drug in combination therapy (e.g., with traditional chemotherapy: doxorubicin and cyclophosphamide or radiation) [50]. Immunotherapies are increasingly being used, with substantial research and development, but they also have some side-effects [51]. For example, immune checkpoints inhibitors (ICI) have been associated with cardiac-related complications including myocarditis, pericarditis, vasculitis, and arrhythmias, occurring in about 1% of patients [52].

The above physiological contexts and CV pathologies are examples of the adverse health effects of many existing and emerging cancer therapies on CV health, which exacerbate the already high CVD morbidity and mortality [11]. For example, CCSs are at significant elevated risk of atherosclerosis and coronary artery disease (CAD) [38], and CVD is their most common cause of death [38,53]. CVD is also the leading cause of death among BC survivors [29], posing a greater mortality threat than BC itself [20], particularly during the 7 year window of opportunity after cancer diagnosis [8]. CVD is also the number one noncancer cause of death among BC survivors (≥50 years), accounting for 35% of non-BC mortality in 2004 [54]. 

## 2. Inflammation and Cardiovascular Disease

Inflammation is essential to the pathogenesis of atherosclerosis, a major pathology of ischemic heart disease (IHD), and the most common cause of HF [55]. There are two types of inflammation—acute and chronic [56,57]. Acute inflammation is a short-lived normal response of a living tissue to injury, trauma, or infection [57,58]. When there is an injury, the body’s immune system releases white blood cells to surround and protect the injured area and speed up the healing process [56]. Characterized by an increase in hepatic synthesis of positive acute-phase proteins such as C-reactive protein (CRP), serum amyloid A, and haptoglobin, acute inflammation is protective to the body and essential in maintaining homeostasis [59]. However, if factors triggering acute inflammation are not resolved, the immune system continues to produce white blood cells and chemical messengers that prolong the process, leading to chronic inflammation [56]. 

Neutrophils are the main biomarkers in acute inflammation, while mononuclear cells (i.e., lymphocytes, macrophages, and plasma cells) participate in the chronic inflammation process. Outcomes of acute inflammation are not severe, such as abscesses or ulcers, while those of chronic inflammation are severe, including tissue destruction, fibrosis, and necrosis [57]. Inflammation process is mediated by proinflammatory cytokines such as interleukin-1 (IL-1), tumor necrosis factor alpha (TNF-α), and interleukin-6 (IL-6), which activate inflammatory cells [57]. Cytokines are small secreted proteins released by cells with specific effects on the interactions and communications between cells [60]. Activated leukocytes secrete at least 15 different low-molecular-weight cytokines and trigger an acute-phase response, which manifests with fever, leukocytosis, increased synthesis of adrenocorticotropic hormones, and production of various acute-phase proteins [57].

Inflammation is regulated by pro/anti-inflammatory cytokines that accelerate or decelerate its pathogenesis. Cytokines either directly or indirectly control inflammatory reactions through their ability to activate or deactivate synthesis of some cellular adhesion molecules [61]. Proinflammatory cytokines are immunoregulatory cytokines that favor inflammation, including IL-1α, IL-1β, IL-6, and TNF-α, responsible for early responses [60]. There exist some chemokines that chemoattract several inflammatory cells and function as proinflammatory mediators, such as LIF, IFN-γ, OSM, CNTF, TGF-β, GM-CSF, IL-11, IL-12, IL-17, IL-18, and IL-8, and that upregulate the production of IL-1α, IL-1β, IL-6, and TNF-α [61]. On the other hand, *anti-inflammatory* cytokines neutralize various aspects of inflammation, including cell stimulation or synthesis of proinflammatory cytokines, thus controlling the magnitude of inflammatory responses/reactions in vivo [61]. Major anti-inflammatory cytokines include IL-4, IL-10, and IL-13. Some anti-inflammatory mediators also act by inhibiting the synthesis of proinflammatory cytokines or by neutralizing/balancing many biological mechanisms of proinflammatory mediators [61]. They include soluble receptors for TNF or IL-6, as well as IL-16, IFN-α, TGF-β, IL-1ra, and G-CSF [61]. 

Studies have associated chronic inflammatory diseases (CIDs) and atherosclerosis [62]. Atherosclerosis is an inflammation of blood vessels [63]. Its pathogenesis starts by fatty deposits inside the lining of artery walls, narrowing the arterial cavity over time (stenosis), which can partially or totally block blood flow, leading to ischemic attack [64,65]. Chronic low-grade inflammation has been linked with CVD risk [66] and a higher risk of cancer [67]. In a large cohort study, higher levels of inflammatory biomarkers were associated with higher HF risk among CID patients than controls matched in terms of age, sex, insurance status, baseline year, and baseline presence or absence of HTN and/or diabetes [62]. CVD has been linked to several inflammatory biomarkers including hsCRP and IL-6 [68], cortisol [69], growth differentiation factor-15 (GDF-15) [70], fibrinogen [71], uric acid [72], and Toll-like receptors (TLR) [73].

CRP mediates the atherothrombosis pathophysiological pathway by increasing plasminogen activation inhibitor 1 (PAI-1) expression and activity in aortic endothelial cells [74]. This pathophysiological process increases atherothrombosis risk among highly susceptible populations, including cancer survivors. In a prospective, nested case–control study with a sample of postmenopausal women with no history of CVD or cancer, hsCRP and IL-6 predicted CV events [68]. In that study, each new CV event case (defined as death from CAD, nonfatal MI or stroke, or a coronary-revascularization procedure), was matched to two controls on the basis of age and smoking status. The study included 122 cases and 244 controls [68]. Independent of all covariates adjusted for, both hsCRP and IL-6 were established as significant predictors of CV events [68].

Cortisol has also been associated with negative CVD outcomes. In a study with a sample of 1881 Korean adults >20 years old, higher levels of cortisol were associated with increased risk for CVD after adjusting for age, BMI, and overall adiposity level [75]. Additionally, findings from a series of prospective cohort studies and random-effects meta-analysis suggested that elevated morning cortisol is a causal risk factor for CVD [69]. 

GDF-15 is associated with cardiac and vascular dysfunction [76]. In a prospective cohort study with 1391 participants (mean age 70 years, no history of CVD, followed for 11 years), GDF-15 was a predictor of CV mortality, after adjusting for CVD risk factors [70]. In another prospective cohort study with 1016 elderly Swedish individuals, GDF-15 was a predictor of CV events during a 10 year follow-up [77]. Additionally, an increase in GDF-15 levels has been associated with a greater likelihood of adverse outcomes in patients admitted for acute HF or renal failure rehospitalizations and/or subsequent CV death [78].

Fibrinogen is another known contributor of atherogenesis, endothelial injury, and thrombogenesis [71]. A meta-analysis review associated fibrinogen with adverse CVD outcomes [71]. In this study, prospective cohort studies with baseline information on fibrinogen levels and details of subsequent major vascular morbidity and/or cause-specific mortality during at least 1 year of follow-up were included, while studies with participants who had a previous history of CVD were excluded. Eligible individual records for 154,211 participants in 31 studies were identified. During 1.38 million person-years of follow-up, 6944 first nonfatal MI or stroke events and 13,210 deaths occurred. The authors found associations between fibrinogen level and risk of coronary heart disease (CHD), stroke, other vascular mortality, and nonvascular mortality in healthy middle-aged adults [71]. 

Uric acid, the end-product of purine metabolism and an inflammatory biomarker, is also a correlate of adverse CV outcomes [72]. Studies have associated serum uric acid (SUA) and negative CVD outcomes including increased CVD risk [79] and CVD mortality [80]. In a meta-analysis of prospective observational studies, SUA was associated with CVD and/or all-cause mortality [80]. Studies included in this meta-analysis assessed baseline SUA levels and subsequent CV or all-cause mortality events in the general population. At a follow-up time greater than 4 years, baseline SUA level independently predicted future CV mortality [80].

Similarly, TLRs are transmembrane pattern recognition receptors (PRRs) that have a critical role in innate immune response, inflammation, immune cell regulation, cell survival, and proliferation [81,82,83]. Activated by both pathogen-associated molecular patterns (PAMPs) and damage-associated molecular patterns (DAMPs), TLRs are involved in activating inflammatory cascades and subsequent neuroprotective or harmful effects on CVDs [82,83,84]. In rodent studies, the TLR9 connection is well defined, but few human studies have connected TLRs to CVD; moreover, to the best of our knowledge, none has associated TLR signaling with chemotoxicity [85]. All 13 mouse TLRs (TLR1 through TLR13) are recognized and have distinct roles in atherosclerosis pathological processes [86], atherothrombotic CVD [82], cerebral vascular diseases, acute ischemic stroke, intracerebral hemorrhage, cerebral venous sinus thrombosis [87], cardiac dysfunction, and HF [88].

## 3. Inflammation and Cancer

Inflammation has been linked with increased risk of several types of cancer [89], including BC [90] and its progression [91,92]. BC survivors have higher levels of proinflammatory biomarkers, particularly during the first 5 years following cancer diagnosis [92]. Some of those increased biomarkers include CRP, TNF-α, and IL-6 [93]. Among BC survivors, inflammation has been associated with adverse health outcomes including cognitive impairment and lower cognitive performance, fatigue, depression, and poorer quality of life (QoL) before, during, and after cancer treatment [93,94]. 

Among cancer survivors, inflammation is exacerbated by systemic treatments. Chemotherapy, for example, has been associated with increased levels of inflammatory biomarkers such as TNF receptor II (TNF-RII), which has also been associated with post-chemotherapy fatigue among BC survivors [95]. In a case–control study, a poorer inflammation profile was established among chemotherapy-treated BC survivors than matched controls [96]. In this study, 19 biomarkers including TNF super family member 13b (TNFSF13B), GDF-15, peptidase inhibitor 3, insulin growth factor-binding protein 7, proprotein convertase subtilisin/kexin type 9 (PCSK9), osteopontin, and perlecan were all associated with chemotherapy [96]. Inflammation was associated with cardiac dysfunction, observed through independent associations with lower left-ventricular ejection fraction [96]. 

## 4. Anti-Inflammatory Pharmacological Interventions against MACE and Need for Other Innovative Interventions

There is a well-documented role of inflammation in MACE pathogenesis, and inflammation has been a major treatment target. Several anti-inflammatory drugs are being clinically used, not only to treat inflammation, but also to reduce risk of CVD [16,97], although findings have been mixed depending on specific characteristics of the population of interest [98]. For example, methotrexate (MTX) is a disease-modifying antirheumatic drug used for treatment of chronic inflammatory disorders [16] and a first line-therapy drug for patients with rheumatoid arthritis (RA) [16,97]. MTX has been associated with reduced risk of CVD events in patients with RA [16,97], while it did not show any inflammatory or CVD risk reduction benefits among patients with stable atherosclerosis [14]. In a meta-analysis systematic review of MTX use and risk for CVD, MTX was associated with a 21% lower risk of total CVD and an 18% lower risk of MI [16]. However, the CVD and inflammatory risk reduction of MTX was not consistent in a randomized, double-blind CIRT trial [14]. The CIRT trial enrolled 4786 patients with previous MI or multivessel coronary disease who also had either type 2 diabetes or metabolic syndrome, but did not have RA or any other inflammatory disease [14]. The CIRT trial had to be ended sooner than anticipated because low dose of MTX did not reduce levels of IL-1β, IL-6, or CRP and did not result in fewer CV events than placebo [14].

Two important anti-inflammatory drugs that have been associated with reduced risk of CVD are colchicine in the COLCOT trial [18] and canakinumab in the CANTOS trial [15]. Similar to the CIRT trial, both COLCOT and CANTOS trials were also randomized, double-blind, placebo-controlled studies and enrolled patients with MI [15,99]. The CANTOS trial enrolled 10,061 patients with a history of MI and residual inflammatory risk defined as hs-CRP levels ≥2 mg/L, with the primary efficacy end point being nonfatal MI, nonfatal stroke, or CV death [15]. The COLCOT trial enrolled 4745 participants with MI within the last 30 days and completion of all intended coronary revascularization, and the primary efficacy endpoint was a composite of CV death, resuscitated cardiac arrest, MI, stroke, or urgent hospitalization for angina requiring coronary revascularization [18].

Unlike MTX in the CIRT trial, both colchicine and canakinumab were associated with reduced risk of MACEs among patients with MI [15,18,99]. Canakinumab significantly reduced the rate of recurrent CV events by 15% (HR = 0.85; 95% CI = 0.74 to 0.98) compared to placebo, independent of lipid-level lowering medications [15]. Anti-inflammatory effects of canakinumab were also found in a double-blind, multinational phase IIb trial of 556 patients with well-controlled diabetes mellitus and high CV risk, although no major effect was observed on low-density lipoprotein cholesterol or high-density lipoprotein cholesterol [100]. Although canakinumab has high efficacy, it is an expensive monoclonal antibody, and its high cost can be prohibitive for many patients, especially poorer individuals with higher copay and limited access to comprehensive drug insurance coverage. In addition to the high and access-prohibitive cost, canakinumab was also associated with higher incidence of fatal infection than placebo in the CANTOS trial [15]. 

The mixed findings from the above pharmacological trials against inflammation and CVD, along with the concurrent high burden of both CVD and cancer, and a disproportionately high burden for cancer survivors, represent a strong basis for the need for innovative approaches to CVD prevention, aimed at improving treatment outcomes and QoL during cancer survivorship. Neighborhood-based interventions such as greenspaces are low-risk and population-based strategies that should be incorporated into this innovation. Greenspaces are great assets, and they have been proposed by scholars as potential avenues for increasing community and individual resilience from multiple public health threats including the ongoing COVID-19 pandemic, climate change, structural racism, and the burden and inequity of persistent chronic diseases, including CVD and cancer [17,101]. The well-known health benefits of greenspaces can be harnessed in reducing CVD risk and preventing cancer treatment-induced cardiotoxicities, improving inflammatory profile and CV health outcomes, reducing health disparities, and improving social and health equity.

## 5. Greenspace Interventions

### 5.1. Greenspace and Health Outcomes 

The US Environment Protection Agency (EPA) defines greenspace as any land partially or completely covered with grass, trees, shrubs, or other vegetation [102]. Three systematic reviews, one in 2018 and two in 2021, found that greenspaces have beneficial effects on physical and mental health and wellbeing [103,104,105]. Greenspace has also been associated with improved mental health in children and adults [106], as well as physical and socioemotional wellbeing [107]. 

The impact of greenspace on mental health is well researched, and evidence has been presented with different outcome measures including reduced likelihood of depressive symptoms [108], reduced stress [108], lower levels of anxiety symptoms [108], improved cognitive functioning [109], improved psychophysiological stress response [110], and improved children’s socioemotional health [107]. In addition to a positive impact on mental health, greenspace’s physical health benefits have been measured with different outcomes including children’s physical activity (PA) [107] and improvement in cancer-related outcomes across the cancer control continuum through different mediating factors (i.e., higher PA, reduced air pollution, improved psychological factors, and improved social environment) [111]. 

### 5.2. Greenspace and Biopsychosocial Plausible Pathways to Positive Health Outcomes 

While underlying mechanisms linking greenspace and positive health outcomes remain complex and partially understood, research has focused on different pathways including environmental factors, physiological and psychological states factors, and behavioral factors [112]. Kuo (2015) critically appraised 21 plausible causal pathways, each one having been empirically linked to nature or greenspace by controlling for important confounders [112]. Kuo (2015) suggested that enhanced immune functioning is a central pathway linking greenspace and health [112]. In the environmental pathway, Kuo appraised the role of chemical and biological agents, naturally released by plants, including antimicrobial volatile organic compounds (i.e., phytoncides) [112]. Phytoncides have many known health benefits including reducing blood pressure (BP), altering autonomic activity, and boosting immune functioning [113]. In the physiological and psychological states, Kuo appraised different pathways, including the post-nature-exposure increase in protein-hormone adiponectin [114], which has protective effects on CVD risk [115] and boosts immune system functioning [112]. In the behavioral pathway, Kuo appraised links between nature contact and PA, obesity, sleep, and social ties, suggesting that these associations might be mediated by an increase in adiponectin post nature exposure [112]. Kuo applied three criteria in the determination of a particular pathway’s centrality and suggested that “enhanced immune functioning” is a central pathway in the relationship between nature and health [112].

### 5.3. Greenspace and CV Health

Greenspace has been associated with favorable CV outcomes including increased angiogenic capacity [116], reduced CVD risks [117,118], and decreased CVD morbidity and all-cause and CV mortality [104,119]. Those associations have been established through different hierarchical levels of evidence, including from ecological studies to multilevel studies and experimental studies. 

#### 5.3.1. Ecological Studies of Greenspace and CV Health

Numerous ecological studies have linked greenspace and CV health. One study in Belgium, for example, used a proxy measure of CVD medication sales to assess links between greenspace and CV health [120]. In this study, a negative correlation between residential greenspace and CVD medication sales was found for the 2006 to 2014 time period [120]. In analyzing data for 11,575 census tracts, the authors observed a correlation between greenspace and a reduction in CVD medical expenditure. This association was even stronger in regions with lower greenspace cover and was observed for only specific ranges of greenery, suggesting a threshold (necessary dose) and a maximum amount of greenery for positive outcomes or a ceiling effect, in pharmacological terms [120]. Similar conclusions were made in another study with neighborhood-level greenspace measures and different measures of CV health in China [121]. With a neighborhood-level representative sample, Leng et al. (2020) examined links between neighborhood-level greenspace and CV outcomes and found positive associations [121]. Residents in neighborhoods with a *greenspace ratio* lower than 28% or *green view index* lower than 15% had higher risk of physical inactivity, overweight or obesity, HTN, and stroke [121]. In another study in Brazil, exposure to greenspace was associated with reduced CV mortality [122]. In this study, census tracts were used as the unit of analysis, and data from deaths due to IHD and cerebrovascular diseases among residents (≥30 years) from 2010 to 2012 were used. A protective effect of greenspaces on IHD mortality was observed among the greenest sectors of all strata and was higher for those of a lower socioeconomic status (SES) [122]. The main limitation of such ecological studies is that observed relationships might be due to some unknown confounding factor that was not measured in the study. To reduce this study design limitation, other study designs such as multilevel studies, which use multiple levels of analytical units, are necessary. 

#### 5.3.2. Multilevel Studies of Greenspace and CV Health

Several multilevel studies have linked greenspace and CV health. For example, a cross-sectional study compared two cities with merged data from the Barcelona Health Interview Survey (2016) in Spain and the Belgian Health Interview Survey (2013) [123]. Using distance to nearest greenspace as a proxy measure for residential greenness exposure and assessing CV with self-reported medication use for MI, HTN, and CV conditions, the authors found that one interquartile range increase in distance to nearest greenspace was associated with increased risk of HTN (OR: 1.15; 95% CI: 1.04–1.26) and use of CV medication (OR: 1.15; 95% CI: 1.04–1.27) [123]. 

Other large-scale epidemiological studies, including two prospective cohorts of 1.3 million people (aged ≥19 years), followed from 2001 to 2011 (10 years) in Canada [124], and 1.26 million subjects (aged ≥30 years), followed from 2001 to 2013 (12 years) in Italy [125], consistently found that living near greenness profoundly affects both CV and all-cause mortality. In an Australian study of adults (aged ≥45 years), an increase in tree canopy was associated with lower odds of CVD prevalence [126]. In another study in the UK, with data from the European Prospective Investigation of Cancer Norfolk UK cohort (*n* = 24,420), residential neighborhood greenspace was associated with reduced CVD risk [127]. A Korean study with data from the National Health Insurance Service National Sample Cohort (*n* = 351,409, aged ≥20 years) also linked greenspace amount to lower CVD [128]. In Lithuania, associations between both distance to and use of urban greenspaces and prevalence of CVD and its risk factors were investigated with data from a population-based Kaunas cohort study (*n* = 5112, 45–72 years old, free from CVD at baseline) [129]. In this study, greenspace use was associated with better CV health outcomes [129].

#### 5.3.3. Experimental Studies of Greenspace and CV Health

Fewer investigators have used experimental design in studying greenspace and CV health. One of those used a field experiment looking at acute effects of visit to urban green environments on CV physiology [130], while another one used a natural experiment studying loss of trees and CV risk factors [131]. Both studies found positive relationships between greenspace and CV health. In the field experiment, 36 female volunteers visited three different types of urban environments for 45 min (urban forest, urban park, and a built-up city center) in Finland. Overall, 15 min was spent during sedentary viewing, while 30 min was spent walking [130]. The authors found that visits to greener environments were associated with lower BP, lower heart rate, and higher indices of HRV. In the second study, a natural experiment evaluated trees lost due to an invasive forest pest, the emerald ash borer [132]. The authors found that women living in a county infested with emerald ash borer had increased CVD risk (HR = 1.25, 95% CI: 1.20–1.31) [131]. The health benefits of walking in a forest with respect to CV relaxation were evaluated with a sample of 48 young adult males and showed CV relaxation effects including reduced heart rate, BP, and anxiety, as well as improved HRV and mood states [133]. Walking in a forest environment promotes CV relaxation by facilitating the parasympathetic nervous system and by suppressing the sympathetic nervous system, while reducing negative psychological symptoms [133]. Lastly, a recent systematic review and meta-analysis consistently linked greenspace and positive CV outcomes, including reductions in heart rate by 3.47 (95% CI: −4.04, −2.90), diastolic BP by 1.97 (95% CI: −3.45, −0.49), HDL by 0.03 (95% CI: −0.05, <−0.01), and CV mortality by 0.84 (95% CI: 0.76, 0.93) [104].

In summary, from ecological studies to multilevel studies, from experimental studies to few systematic reviews, there is consistent evidence for the role of greenspace in improving CV health, from reducing CVD risk factors to lowering the incidence of CVD and reducing CVD-related mortality. Greenspace interventions with a focus on their usage might indeed be a promising venue in CVD prevention and CVD clinical management among cancer survivors.

### 5.4. Greenspace and Inflammation

Greenspace has been associated with an improved inflammatory profile. For example, forest bathing, a traditional Japanese nature immersion practice “Shinrin-Yoku”, has been linked with reductions in inflammation and stress [134]. Forest bathing involves a walk in a forest aimed at integrating and harmonizing humans with nature by using all human five senses (sight, hearing, taste, touch, and smell) [135]. The anti-inflammatory benefits of forest bathing were evident in two different experimental studies [134,136]. In a study with a group of chronic obstructive pulmonary disease patients [134], significant decreases in proinflammatory cytokines, including IFN-γ, IL-6, IL-8, IL-1β, TNF-α, and CRP were observed among exposed groups [134]. Similarly, a study of 24 randomly assigned patients with HTN (60–75 years) compared to a normotensive control group (*n* = 12 per group) found that the forest group had significant decreases in systolic blood pressure (SBP) and diastolic blood pressure (DBP) [136]. In this study, the experimental group visited a broad-leaved evergreen forest to experience a 7 day and 7 night trip, while the control group went to a city area for a 7 day and 7 night trip. Both groups had similar day activities, sleep schedules, meals, and hotel environments. Blood samples were collected before and after exposure, and there was no difference in all biomarkers investigated at baseline. After the experiment, the forest group had a significant decrease in SBP, DBP, and IL-6, although no difference was observed for TNF-α [136].

In another study, a crossover design was used to investigate the effect of 2 h exposure to forest or urban environments on cytokines, antioxidants, and stress levels in young adults [137]. Subjects were assigned to each group on the basis of demographic characteristics. One group was first exposed to a forest environment, while the other group was first exposed to an urban environment. Carryover effects were avoided by moving participants to a small town in a rural area for an equal amount of time. For both groups, blood samples were collected, and serum cytokine levels were assessed for IL-6, IL-8, TNF-α, and glutathione peroxidase (GPx). The authors found that serum IL-8 and TNF-α levels significantly decreased after exposure to a forest environment [137]. 

In a study of college students in China, greenspace was linked to reduced inflammation and stress [138]. Twenty students were randomly assigned to either control or experimental groups (*n* = 10 each) [138]. The experimental group went for a 2 night trip in a forest, while the control group went to a city, while controlling important covariates. Serum cytokines were measured, and no baseline differences were observed. After the experiment, the experimental group had decreased levels of malondialdehyde (MDA), IL-6, TNF-α, and cortisol [138]. Similar observations on stress reduction were noted in two other studies, one looking at cortisol [139] and another looking at adrenaline and noradrenaline [140]. Other experimental studies have examined the impact of forest bathing on the immune system and found an increase in natural killer (NK) cell activity and expression of anticancer proteins [141,142]. Table 1 summarizes some of the studies that have looked at greenspace and different biomarkers. 

The summary of work with studies mostly conducted in Asian settings suggests robust evidence for beneficial effects of greenspace on both inflammation and CV health [151]. However, there is limited evidence for studies on greenspace and inflammatory biomarkers in western countries including the US. Similarly, the available experimental/interventional studies on greenspace and CV health have not been systematically reviewed to assess the level of existing evidence. Additionally, there is limited literature with a focus on cancer survivors, regardless of their increased vulnerabilities. Yet, greenspace benefits might be harnessed in preventive cardio-oncology care. The limited literature on greenspace- and health-focused studies in the US might be due to less funding on the topic, which currently tends to require additional factors such as air pollution, to be funded. Ironically, air pollution is a significant predictor of inflammation [152], and greenspace is a significant contributor to reducing air pollution in urban settings [153], which suggests the need for adding a focus on greenspace in inflammation and urban health studies. This pathway could be leveraged in studying greenspace in the US to reduce inflammation and improve cancer survivorship and survival.

## 6. Proposed Conceptual Framework

Inflammation has a negative impact on CV health and increases CVD risk while exacerbating adverse CVD conditions. Furthermore, biomarkers of inflammation remain underexplored mediators between greenspace and CV health outcomes. Using the socioecological model of health, we propose a new conceptual framework to be adapted in greenspace and cardio-oncology research (Figure 1). The socioecological model is a well-known framework in studying complex levels of influence and an important model in designing multilevel interventions focused on underlying socioeconomic determinants of health [154]. In the proposed model, upstream factors such as actions taken at a policy level (e.g., greenspace friendly policies) influence possibilities at organizational, community, and individual levels (e.g., increase in access and promotion of use of greenspaces). Universal access to free and safe greenspaces enables and/or influences interpersonal positive social interactions and individual-level perception regarding availability, safety, access, and ultimate use of greenspaces, which lead to positive health outcomes, including reduced inflammation, CVD risk, and cancer treatment-induced cardiotoxicity.

In the model, we suggest important confounders that need to be controlled for including individual demographic factors (e.g., age and gender), socioeconomic factors (e.g., race, ethnicity, income, education level, and marital status), behavioral factors (e.g., PA, dietary habits, smoking status, alcohol consumption, social support, and engagement) comorbidity conditions (other chronic diseases), and social and built neighborhood environment characteristics (*pathogenic factors*: air-polluting factories, liquor or tobacco stores, and fast-food restaurants; *salutogenic factors*: grocery stores, churches, and community libraries). A graphical abstract is annexed in Appendix A. 

## 7. Way Forward and Implications for Translational Science and Future Research

Associations between greenspace and numerous positive health outcomes are well documented, including improved physical health, mental health, and overall social wellbeing [104]. CVD and cancer outcomes are both known to be positively impacted by greenspace, and the quality of life and survival of cancer survivors would be boosted by innovative greenspace interventions including nature prescriptions and other post-cancer treatment recommendations that include increased personal exposure to greenspaces and immersion in greenspaces. Additionally, innovative messaging approaches to increase greenspace views from one’s home or work, as well as the importance of tree-lined streets, can offer individual and community-level benefits. Communities should invest in programs that enhance human–nature interactions such as public hang-out events, community greenspace beautification events, or “kids to parks” day events [155].

Furthermore, with awareness of different societal factors that impact human–nature interactions such as increased urbanization and reduced accessibility of natural environments [156], as well as increased screen time on computers at work and/or on TVs or smartphones at home [157], other innovative interventional approaches such as virtual human engagement with nature or virtual reality (VR) [158], or other nature visual stimuli such as nature images or indoor plants might be a viable option for nature exposure. Indeed, the intention of VR was to facilitate human–nature engagement and support humanity in the pursuit to continue enjoying nature’s health benefits through VR [159,160]. Consequently, VR has been associated with positive health outcomes including improved positive mood levels and attention restoration compared to an indoor setting without nature [161,162]. Additionally, a systematic review found that VR interventions are effective in managing symptoms of depression, fatigue, pain, and anxiety [163]. Similarly, exposure to images of nature has been associated with improved health outcomes, including reduced stress [164]. 

Although the magnitude of effects of images of nature or VR on health outcomes might be relatively smaller compared to exposure to a real nature setting [165], they still offer potential opportunities for innovation in clinical practice including use in palliative care [166] and in special settings when access to nature is impaired by weather, disability, or other socioeconomic disadvantages, or when the risk of injury outweighs the health-promoting benefits of real nature [167]. Considering the growing evidence on the role of VR or nature images in health outcomes, it is time to consider either of these potential interventions in health promotion. Such interventions would need to be customized to specific individual needs and available resources, particularly among special-needs population groups, including those with limited access to real nature. 

Future studies on greenspace and cardio-oncology should consider adapting and using this novel proposed model in their conceptual designs and analytical approaches, as we work together to reduce health disparities and improve environmental justice and intergenerational health equity. 

## 8. Conclusions

Greenspace or nature-based interventions and their role in improving inflammation and CV health represent a new and growing area of research. However, there is a dearth of research focused on cancer survivors exploring the scientific premise of greenspaces in improving survivorship and survival. Cancer survivors can benefit from such interventions by improving CV health outcomes, and the role of biomarkers remains under-investigated. There is a need to investigate the mediating effects of inflammation through different biomarkers that can be targeted in reducing cancer treatment-related cardiotoxicity. Such innovations will provide another clinical perspective in managing the highly increasing prevalence of chronic diseases in the US context and beyond. Additionally, there is a need for increased research funding for greenspace interventions from the National Institute of Environmental Health Sciences (NIEHS) and other funding agencies. The novel conceptual framework proposed in this paper could guide this emerging area of research on greenspace and cardio-oncology to improve CV health outcomes and cancer survivors’ quality and length of life. The authors challenge themselves to incorporate the newly proposed framework into their scholarly works and invite other scholars to join them in adapting and customizing the model to other greenspace-focused research or other research innovations focused on neighborhood factors that leverage the socioecological framework.

## Figures and Tables

**Figure 1 ijerph-19-02426-f001:**
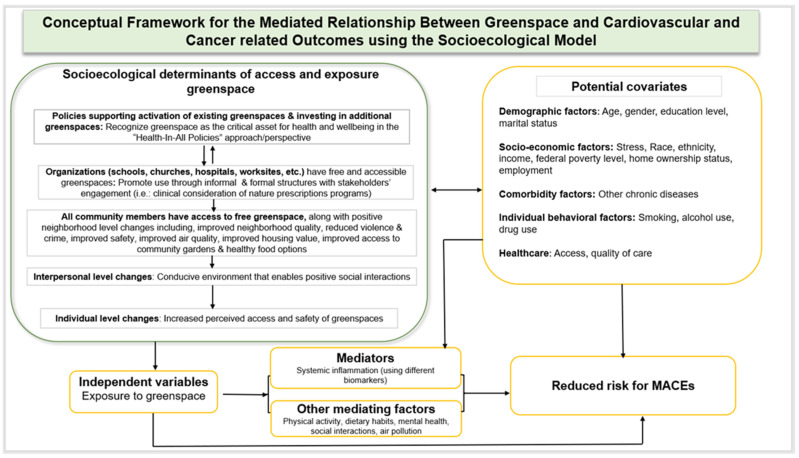
Proposed conceptual framework.

**Table 1 ijerph-19-02426-t001:** Links between greenspace and different biomarkers in humans (samples of studies).

No.	Sample Studies	Greenspace Exposure	Measured Biomarkers and Outcomes Observed
1	Yeager et al. (2018) [116]	Residential greenness	Reduction in oxidative stress biomarkers: (1) F2-isoprostane Reduction in stress biomarkers: (1) Urinary levels of epinephrine No significant change in norepinephrine or other catecholamines and monoamines Improvement in circulating angiogenic cell profile
2	Mao et al. 2012 [138]	Forest bathing	Reduction in pro-inflammatory biomarkers levels: (1) Tumor necrosis factor alpha (TNF-α), (2) Endothelin (ET-1) (3) Interleukin-6 (IL-6) No significant reduction in C-reactive protein (CRP) Reduction in other biomarkers including oxidative stress: (1) Malondialdehyde (MDA) No significant change in total superoxide dismutase (T-SOD) Reduction in stress biomarker: (1) Serum cortisol No significant change in testosterone
3	Demark-Wahnefried et al. 2018 [143]	Vegetable gardening	Significant decrease in telomerase activity No significant change in cortisol and IL-6 Cancer survivorship implication: (1) Improved overall QoL (2) Increased consumption of fruits and vegetables
4	Wu et al. 2020 [144]	Forest bathing	Reduction in proinflammatory biomarker levels: (1) CRP
5	Egorov et al. 2017 [145]	Vegetated land cover near residence	Improvement in all measured biomarkers: (1) Reduced allostatic load (2) Reduced odds of having low levels of norepinephrine, dopamine, and dehydroepiandrosterone (3) Reduced odds of having high levels of epinephrine, fibrinogen, vascular cell adhesion molecule-1, serum IL8, and saliva α-amylase
6	Antonelli et al. 2019 [139]	Forest bathing	Reduction in stress biomarker: (1) Salivary cortisol
7	Mao et al. 2017 [146]	Forest bathing	Reduction in proinflammatory biomarker levels: (1) ET-1 (2) IL-6 No significant change in TNF-α and CRP Reduction in other CVD pathological factors including brain natriuretic peptide (BNP), a biomarker of heart failure, and constituents of the renin angiotensin system (RAS): (1) Renin (2) Angiotensin II (Ang II) (3) Angiotensinogen (AGT) (4) Angiotensin II type 1 receptor (AT1) (5) Angiotensin II type 2 receptor (AT2) No significant change in NT-ProBNP Improvement in oxidative stress biomarkers: (1) Increase in T-SOD (2) Reduction in malondialdehyde (MDA)
8	Mao et al. 2012 [136]	Forest bathing	Reduction in proinflammatory biomarkers: (1) ET-1 (2) IL-6 (3) Homocysteine (Hcy) Reduction in constituents of RAS: (1) AGT (2) AT1 (3) AT2 No significant change in renin and Ang II No significant change in TNF-α
9	Li et al. 2016 [147]	Forest bathing	Increased anti-inflammatory biomarkers: (1) Serum adiponectin Reduction in stress biomarkers: (1) Urinary dopamine (2) Urinary adrenaline
10	Ochiai et al. 2015 [148]	Forest therapy	Reduction in stress biomarkers: (1) Urinary adrenaline (2) Serum cortisol
11	Grazuleviciene et al. 2016 [149]	Green exercise	Reduction in stress biomarkers: (1) Cortisol
12	Park et al. 2017 [150]	Vegetable gardening	Reduction in proinflammatory biomarkers levels: (1) TNF-α No significant change in CVD biomarkers: (1) Blood cholesterol (2) Low-density lipoprotein (LDL)
13	Jia et al. 2016 [134]	Forest bathing	Reduction in proinflammatory biomarker levels: (1) IFN-γ (2) IL-6 (3) IL-8 (4) IL-1β (5) TNF-α (6) CRP

## Data Availability

Not applicable.

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
