# Peer review of "Greenspace, Inflammation, Cardiovascular Health, and Cancer: A Review and Conceptual Framework for Greenspace in Cardio-Oncology Research"

_ijerph, 2022, doi:10.3390/ijerph19042426_

Round 1

Reviewer 1 Report

Thanks for an interesting, clear, and well-structured paper.    This ms identifies an important focus (CV health outcomes in cancer survivors), provides a clear review of: the role of inflammation in both CVD and cancer, the current state of anti-inflammatory pharmacological interventions against MACE (major adverse cardiovascular events), and the literature tying greenspace to health and CV health, specifically.   Intro
  • "it is necessary to incorporate greenspace-related behavioral interventions in overall CVD and cancer care process." what is cancer care process, and if this is important, why does greenspace-related behavioral interventions not appear in the model?
  • There's a plethora of greenspace-health reviews recently; pls describe the closest neighbors of this lit review and how this differs and why the gaps you've filled are important to fill.  (Note: I do appreciate the narrower focus, as it allows you to go into more depth in reviewing the evidence related to the specific concern of cardio-oncology, whereas there seems very little point in constantly publishing greenspace-health reviews that all say roughly the same thing.)
Greenspace lit review
  • Seems strangely incomplete in places; there are far more than three systematic reviews tying greenspace to health!, review includes Egorov 18 (17?) but not 2019, 2020 (the listing and description of so many studies in a Table suggests this is a comprehensive review, even if the necessary qualification is given in the text -- perhaps give the criteria by which some studies were excluded?)
  • In reviewing ecological studies on greenspace-CV health, important to note what is/isn't controlled (income, at least!)--personally, my mind simply reads all those findings as "better income, greener neighborhoods, better CV outcomes." In addition, it might make for a more compelling argument to present the experimental, mechanistic work first, since it would reduce (for me) the tendency to dismiss the ecological studies out of hand.
  • It's by no means necessary, but I found myself wishing for a diagram that would give all the biomarkers tied to CV outcomes and show which of those have been tied to greenspace, which not, and which haven't been tested. I suspect it would be a compelling figure and very helpful for researchers.
  • In your paragraph summarizing biomarker work, it would be much appreciated (although not demanded) for you to address whether the greenspace findings might reasonably be attributed to the relative lack of air pollution in forested vs. urban settings. Certainly some Chinese cities have very high levels of air pollution; I don't know whether that's the case for the urban conditions in Mao's work, or the Korean or Japanese urban conditions, or for US work like Egorov's or Yeager's.
Virtual Nature and Health
  • might be nice to mention the many studies showing that visual-only nature exposure has clear effects, and to point out that images of nature make it possible to provide prolonged exposure. Might also be nice to explain why you advocate the relatively expensive intervention of VR nature over images and views of nature, particularly when socioeconomic disadvantage is so tied to lack of access to nature.
Conceptual framework:
  • why do [Policies supporting] impact only the availability of organizations' greenspaces, and not those of the public?
  • otherwise, the framework seems seems largely fine. I'm unclear on the added value, here, though -- do we need a framework that puts policy-level factors in the same model as mediators? how does this differ from previous frameworks? (I apologize if I am simply coming from the wrong field to appreciate this.)
  • For me, I might have appreciated a text description of specific policy and environmental targets tucked into the model (e.g., everyone has free access to safe green places). (What is "Health-in-All?") If you do write a "recommendations for policy/practice" paragraph, the virtual nature ideas might fit better there than in their currently somewhat awkward spot.
  • For me, a large missing piece here is suggestions for health care practitioners who serve cancer patients. Obviously, you could simply recommend that oncologists could issue "nature prescriptions" along with other post-cancer recommendations-- but I'm guessing that with a bit of thought on how these prescriptions might be adjusted for the specific population of cancer survivors, you might have very interesting and useful suggestions.
  • Also, while you do give the definition of greenspace as any space incorporating vegetation, i would caution that it's likely most readers will understand the recommendations in the model as being about green open space (parks, ballfields, etc) and unfortunately, sprawl, as opposed to tree-lined streets, green yards, window boxes, and other ways in which vegetation can be integrated into even very dense urban fabric. If that's your intention, fine -- but I think the literature strongly points to the importance of views from one's home or work, tree-lined streets and other forms of urban nature which are "at hand" and do not require visiting to experience.
I did not pay attention to copyedit-level concerns, but these two popped out at me.   332: casual -->causal 344: meditated --> mediated    

Author Response

Thank your your thoughtful review of our manuscript. The attached manuscript has been greatly improved due to your constrictive feedback. 

Thank you! 

Reviewer 2 Report

The review entitled ,,Greenspace, Inflammation, Cardiovascular Health and Cancer: A Review and Conceptual Framework for Greenspace in CardioOncology Research’’ is very interesting and could have an impact on the science community. The authors have been shown how greenspaces can reduce inflammation and associate with CV health benefits. They reviewed the state of knowledge on associations among inflammation, CVD, cancer, existing pharmacological interventions. Then they presented the relationship between greenspace and inflammation and discuss greenspace benefits on CV health and necessities for virtual nature. Finally, using the socioecological model of health have been proposed a new conceptual framework to be adapted in greenspace and CardioOncology research.

The cited references are current: 91 out of 164 within the last 5 years.

My remarks and recommendations are as follows:

  1. Page 5, line 216: “CHD” is not defined.
  2. Page 7, line 324: “PA” can be defined here, not in line 343.
  3. In the absence of Figure 1, Figure 2 numbering is not appropriate, please correct it.
  4. There is no reference to “Appendix A” in the text.
  5. There are some formatting error in the References, especially Reference 134 and 138.
  6. I slightly missed some practical recommendations and ways for implementing virtual nature. Please complete this chapter with some practical recommendations.

It can be recommended for publication in Special Issue of International Journal of Environmental Research and Public Health: Evidence for the Salutary Effects of Nature-Based Interventions (NBI) for Clinical and Public Health Practice, however after minor revision.

Author Response

(The authors gave the same response as above.)
